# A systematic scoping review and thematic analysis: How can livestock and poultry movement networks inform disease surveillance and control at the global scale?

Sara C. Sequeira[1☉*], Natalie Sebunia[1‡], Jessica R. Page[2‡], Taiwo Lasisi[3‡], Greg Habing[1‡], Andréia G. Arruda[1‡]

1 Department of Veterinary Preventive Medicine, College of Veterinary Medicine, The Ohio State University, Columbus, Ohio, United States of America, 2 Hodesson Veterinary Medicine Library, The Ohio State University Libraries, Columbus, Ohio, United States of America, 3 18th Avenue Library, The Ohio State University Libraries, Columbus, Ohio, United States of America

☉ These authors contributed equally to this work.
‡ NS, JRP, TL, GH, and AGA also contributed equally to this work.
* sequeira.23@osu.edu

## Abstract

The increasing threat of emerging infectious diseases affecting animal and human populations has prompted closer investigation into how movement-linked interactions contribute to geographic spread of pathogens. Animal movements are a key factor in the spread of diseases like Foot-and-Mouth-Disease and Avian Influenza. Network analysis of animal movement data has become a powerful tool for identifying transmission dynamics and informing disease control. However, a systematic evaluation of its applications across species is lacking. This study addresses this knowledge gap through a systematic evaluation of existing evidence. A modified scoping review was conducted following PRISMA-ScR guidelines and a Population, Index test, and Target condition research structure. Articles published between 1975 and 2024 were retrieved from six databases. Inclusion criteria focused on network analysis research explicitly mentioning livestock and poultry movements. Quantitative analyses in R and thematic analysis in NVIVO provided insights into key network applications. Our review of 203 studies across 52 countries highlighted a steady rise in network-based approaches since 2006, particularly after the 2001 Foot-and-Mouth-Disease outbreak. Cattle (40.3%) were the most studied species, followed by swine (33.2%) and poultry (13.0%). Five themes emerged: network structure, epidemic modeling, targeted control, outbreak analysis, and network inference. These applications demonstrated the flexibility of network analysis in veterinary epidemiology. However, challenges persist due to data accessibility, particularly in low- and middle-income countries. Limited standardized movement data hinder cross-country comparisons and epidemiological insights. Expanding

**Data availability statement:** All RIS files and all exclusions are available at OSF (https://osf.io/vr3cs/files/osfstorage).

**Funding:** There is no specific funding for the scoping review hereby presented but the main authors' research is supported by the Agriculture and Food Research Initiative Competitive Grant no. 2022-68015-36628, awarded by the United States Department of Agriculture. The funders had no role in study design, data collection and analysis, decision to publish, or preparation of the manuscript.

**Competing interests:** The authors have declared that no competing interests exist.

data collection, incorporating weighted connections, and integrating economic and geographic factors could enhance network-techniques. In conclusion, network analysis is a powerful framework for identifying high-risk nodes and designing targeted interventions. Future efforts must improve data standardization, temporal movement dynamics, and incorporate multiple transmission pathways to fully capture the complexity of movement networks and their role in pathogen spread. Moreover, strengthening industry-academic collaborations is crucial for optimizing network-based strategies.

## Introduction

In recent times, the increasing threat of global emerging and reemerging infectious diseases, affecting both animal and human health, has prompted a critical examination of their origins and transmission dynamics [1–3]. Historically, infectious diseases in livestock and poultry populations have demonstrated the potential to trigger large-scale outbreaks and epidemics. The Foot-and-Mouth-Disease (FMD) in the United Kingdom, for instance, resulted in the loss of millions of animals and had significant economic and social impact [4]. Furthermore, Highly Pathogenic Avian Influenza (HPAI) [5], Classical and African Swine Fever (CSF and ASF) [6] and even COVID-19 [7], highlight the ongoing threat posed by infectious diseases to agricultural economies and food security worldwide.

The role of animal movements in pathogen spread and escalation of diseases is well-established in the literature, driving the development of robust livestock and poultry traceability systems worldwide, such as the Cattle Tracing System in Great Britain [8]. With an increasing and broader availability of animal movement data, there has been a surge in studies employing network analysis approaches to characterize and predict dynamics of animal movement networks [9–11]. These approaches are grounded in network theory, which provides a framework for understanding the relationships and interactions between entities in a system. Specifically, network theory focuses on the structure of the network, quantifying the importance of nodes (i.e., individual animals or populations) and their edges (i.e., animal movements), and examining how these elements influence pathogen spread [12]. Network theory offers a variety of metrics that emphasize the significance of nodes and their connections within the network, allowing for a more nuanced understanding of how specific components of the network contribute to pathogen transmission. While individual network-based studies have delved into individual food animal species such as cattle [13,14], swine [15,16], equine [17], small ruminants [18,19] and poultry [20,21], collectively, they offer insights into crucial aspects of disease dynamics. These important dynamic characteristics could be not only used to identify "hot spots" or "super-spreaders" (i.e., key players) for pathogen transmission within these networks but also serve as an excellent source to inform disease surveillance and control in addition to more conventional methods. While some studies have summarized the potential public health benefits of such

approaches [22–24], a significant gap remains in comprehending the general applications of animal movement networks in veterinary preventive medicine.

Systematic reviews on this topic are scarce. One available piece of work focused on a review of the different network-based modelling frameworks used to represent animal movements [25]. Additionally, a study by Martínez-López [26] provided a general overview of the potential uses and limitations of these methodologies for studying animal diseases, though it was not a systematic review. While these studies provide valuable insights and emphasize the importance of these concepts, neither review comprehensively addressed the broader applications of animal movement networks in understanding disease dynamics. Furthermore, they did not differentiate among species or considered the specificities of such studies, limiting the ability to generalize findings across different animal populations and transportation systems. This review's scope extends beyond the focus of previous reviews, aiming to build upon previous studies by systematically addressing a wider array of objectives including:

(i)   Identifying key milestones and developments in the use of network analysis and comparing its applications across livestock and poultry movement studies.

(ii)  Investigating the underlying motivations behind the adoption of network analysis approaches within existing studies.

(iii) Evaluating the strengths and challenges of network analysis in studying disease transmission within livestock and poultry populations.

By comprehensively synthesizing existing evidence using a mixed-methods approach, this review aims to provide insights into how these approaches can be harnessed to enhance veterinary preventive strategies and foster research opportunities in the field.

## Materials and methods

A modified scoping review was conducted following the Preferred Reporting Items for Systematic Reviews and Meta-Analysis for Scoping Reviews (PRISMA-ScR) guidelines [27]. This review's protocol is registered within the Open Science Framework (OSF) platform (registration ID: https://osf.io/g46vq [28]) and can be consulted at Systematic Reviews for Animals & Food (SYREAF) [29]. Search terms were developed based on Population, Index test, and Target condition (PIT) framework [30]. The *Population* was defined as livestock and poultry population across all geographical regions. The *Index test* focused specifically on the application of network analysis, a methodological approach that examines patterns and dynamics of livestock and poultry movements. The *Target condition* was the application of these network analysis in the context of communicable infectious disease surveillance and control. These key elements guided the development of precise search terms to capture literature relevant to the study.

### Eligibility criteria

The inclusion and exclusion criteria for this study were agreed by all authors and can be consulted at OSF Registries [28]. All full-text research conducted in any year, both peer-reviewed and grey literature, were included in this study. Selection criteria included studies explicitly addressing animal populations, more specifically livestock and poultry populations. The research papers had to use or provide insights into the applications of network analysis on studying disease dynamics, by applying these techniques to animal movement data, whether involving transportation by vehicle, grazing, or other forms of relocation of livestock and poultry between facilities. Other animal populations (e.g., wildlife) and other network analysis types (e.g., genomic or molecular data) could be present with the necessary condition of livestock and poultry movement data to be analyzed as part of the effort. Background articles that did not explicitly focus on animal movement network analysis for studying disease transmission dynamics were excluded. The classification of studies as background articles were based on the researchers' description of study design and methodology. Finally, research that was not published in English, Portuguese or Spanish were excluded.

## Information sources

Articles published between January-1975 and February-2024 were retrieved from electronic searches on Web of Science Core Collection (via Clarivate), MEDLINE® (via Clarivate), CAB Abstracts (via Clarivate), Scopus (via Elsevier), Agricola (via EBSCOhost) and ProQuest Dissertations & Theses Citation Index (via Clarivate) databases.

## Systematic search strategy

In order to obtain the highest recall or coverage as possible (i.e., number of relevant citations retrieved divided by the total number of relevant citations), a more general approach including a large number of animal species was first adopted. The benefit to higher recall, as opposed to a higher precision, is to reduce the likelihood of missing data in these types of studies, even though the time required to examine all the data might be longer. Within each element of the PIT framework, search terms and subject headings were combined using Boolean 'OR' operators and different elements were linked using 'AND' operators to ensure comprehensive coverage of relevant topics. To refine the search across databases, truncations and proximity searches were adjusted using the relevant syntax for each database. The search strategy for each of the above-mentioned databases can be consulted at OSF Registries [28].

## Data screening and extraction strategy

Search results were downloaded in Research Information Systems (RIS) format into the bibliographic software Zotero [31] to add any potential missing information. Then, all documents were imported into Covidence [32], an online systematic review software that facilitates the review process and de-duplication. Following de-duplication, title and abstract screening as well as full-text screening were performed by one of the researchers, taking into consideration the previously described inclusion and exclusion criteria.

Data extraction was performed by two of the authors (S.C.S., N.S.). All relevant data from the included full texts was included into a data extraction form using Microsoft Excel [33]. To ensure accuracy and reliability of the data collection, the authors utilized five example papers for training and calibration purposes prior to data extraction. Data extracted included study details (such as author(s), publication year, journal, etc.), population characteristics (e.g., study design, animal population(s) studied, geographical location(s) studied, study period(s) and other descriptive variables), methodology followed (i.e., network analysis details and measures implemented), and strengths and challenges encountered.

## Data analysis and synthesis of results

The data analysis for this scoping review comprised a mixed-approach methods, combining both quantitative and qualitative techniques to comprehensively address the research questions.

For research question (i), focused on the key milestones and applications of network analysis across studies, descriptive statistical methods were employed using R software. Various data visualization techniques were employed, including histograms, scatter plots and maps. Tables were generated reporting frequencies, percentages, means and standard deviations (SD) and/or medians and interquartile ranges (IQR). Maps were created using *sf* [34], and *maps* [35] packages in R.

For research questions (ii) and (iii), which explored the underlying motivations, strengths and challenges within the context of animal movement networks, a qualitative approach was employed using thematic analysis as described by Braun and Clarke [36]. This method provided a structured yet flexible framework for interpreting patterns within a large dataset [37]. The first step included familiarization with the data. Key excerpts, such as authors' rationales for using network analysis and descriptions of challenges, were identified and compiled into an Excel file. These excerpts were uploaded into NVIVO [38], where data was coded inductively to allow codes to emerge directly from the data. This data-driven, or "bottom-up", approach avoided reliance on a pre-defined framework. Codes were systematically reviewed and grouped

into broader themes by identifying shared meanings or commonalities. These themes were refined and integrated into final motivation and challenges' themes, to ensure coherence both within individual themes and across the dataset. Findings from the qualitative analysis were then integrated with the quantitative data to provide a more nuanced understanding of the application of network analysis to animal movement data.

This synthesis allowed for the identification of emerging trends and key areas where further exploration could contribute to the development of veterinary preventive strategies. The combined findings were summarized and discussed in the results section, providing insights into how network analysis can be further leveraged to enhance disease surveillance and control in livestock and poultry populations.

## Results

### Screening process

The initial database search identified a total of 2,604 publications, of which 1,350 (51.8%) were unique after removing duplicates (Fig 1). Title and abstract screening resulted in the exclusion of 70.8% of the studies (956 publications), leaving 394 publications for full-text review. Following a comprehensive evaluation based on inclusion and exclusion criteria, 203 studies were ultimately included for detailed analysis. Most of the final exclusions (n = 191) were due to reasons such as network analysis types other than focused on animal movements (n = 83, 43.7%), background articles that did not explore specific network measures (n = 40, 21.1%) or implications for disease surveillance and control, studies focused on the wrong population (n = 33, 17.4%) and other exclusions (n = 37, 17.9%). All exclusions and the final dataset are publicly available [39] and can be downloaded a comma-separated values file for further analysis.

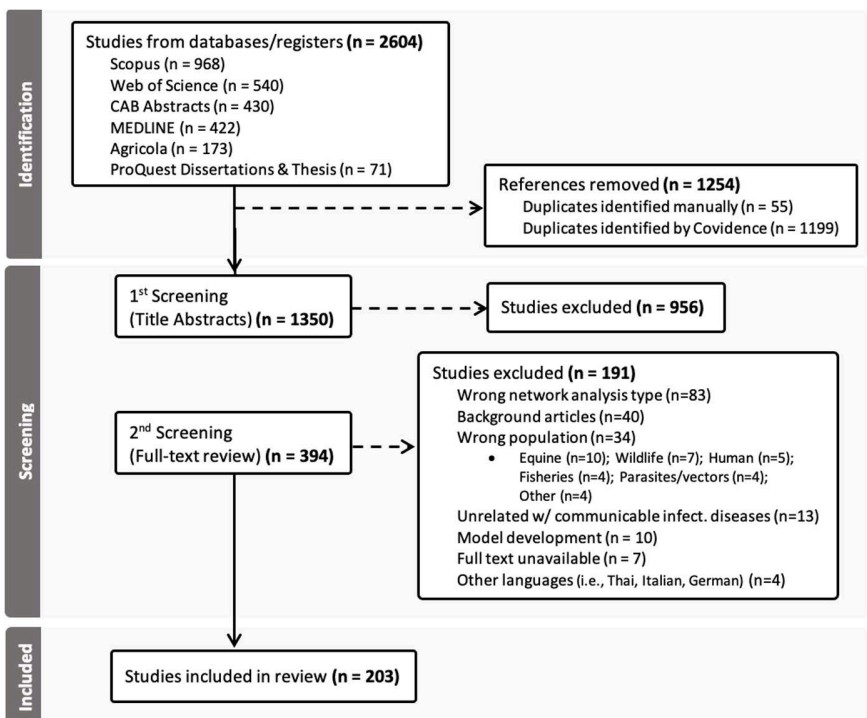

**Fig 1. PRISMA flow diagram of the study selection process.**

## Key milestones and applications of network analysis

**Evolution and key milestones.** The included studies focused on different livestock and poultry populations, with an average number of studies published per year of 11 (SD = 7). They were published from 2006 to 2023, covering animal movement networks in a diverse array of countries and regions. Of the studies, 196 were peer-reviewed journal publications, six were theses (including three Ph.D., one post-graduation in epidemiology, and one master's degree), and one was a conference paper. The peer-reviewed articles were published across of 42 different journals, with the most frequent being *Preventive Veterinary Medicine* (63 articles), followed by *Transboundary and Emerging Diseases* (25 articles), *PloS One* (21 articles), and *Frontiers in Veterinary Medicine* (15 articles).

The publication trend over time is illustrated in Fig 2. The number of studies published per year has generally increased, with notable peaks since 2011 and particularly in 2020. This trend highlights a growing interest in animal movement networks and their application in understanding disease dynamics. The cumulative number of studies also shows a steady rise, indicating an expanding body of literature in this field.

Most studies focused on a single country (195, 96.1%), reaching a total of 52 different countries explored in the scope of this review. Fig 3 illustrates the global distribution of these studies. The United States was the most studied country, with 26 different studies focusing on animal movement networks. The other top four countries studied included France (19 studies), Great Britain countries (17 studies), China (12 studies) and Italy (11 studies). A total of 113 studies (55.7%) focused on specific regions within the country studied. When including multiple countries, the most common was to investigate animal movements within and/or between two countries or including various country-members and territories. One study included 193 country-members of the United Nations [40].

Twenty studies (9.9%) included mixed animal species, with up to five different animal species and most of those were published after 2018 (17, 85.0%). The results highlight cattle as the most studied species (102, 40.3%), followed by swine (84, 33.2%) and poultry (33, 13.0%). The breakdown of species investigated in the five countries with the highest number of studies is summarized in Table 1 and its evolution through time represented in Fig 4. The analysis revealed that studies involving cattle and swine have been the most prevalent throughout the investigated period from 2006 to 2023, with peaks in publication activity observed between 2014 and 2020 for these species. The number of studies on poultry and small ruminants have kept similar numbers across the years, with recent increases in the number of studies for small ruminants but still lower compared to cattle and swine. The vast majority of studies (n = 181, 89.2%) were conducted in high-income (n = 133) or upper-middle-income countries (n = 48), with Europe and the Americas being the most represented continents (n = 124, 61.1%).

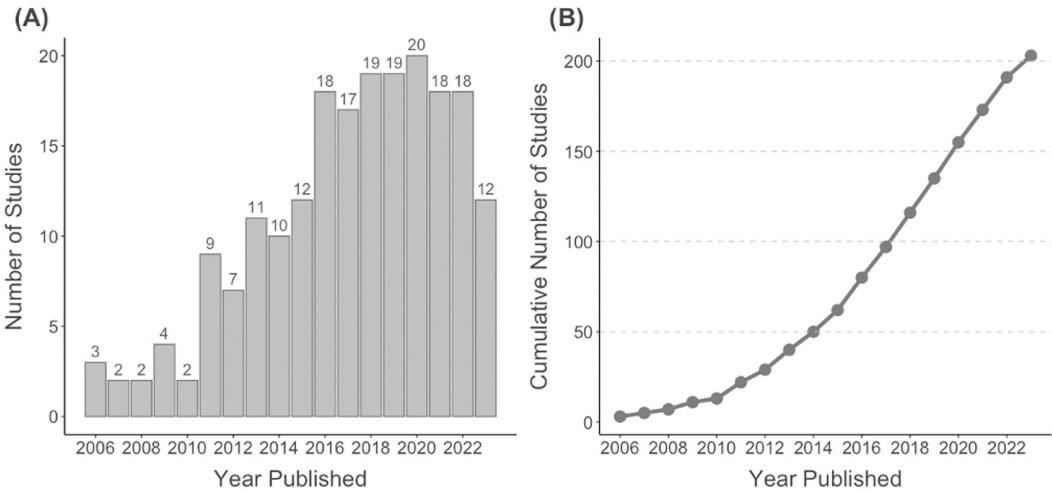

**Fig 2. Trends in research publications over time.** (A) Number of studies published per year from 2006 to 2023. (B) Cumulative number of studies over the same period, showcasing the steady growth in the body of literature.

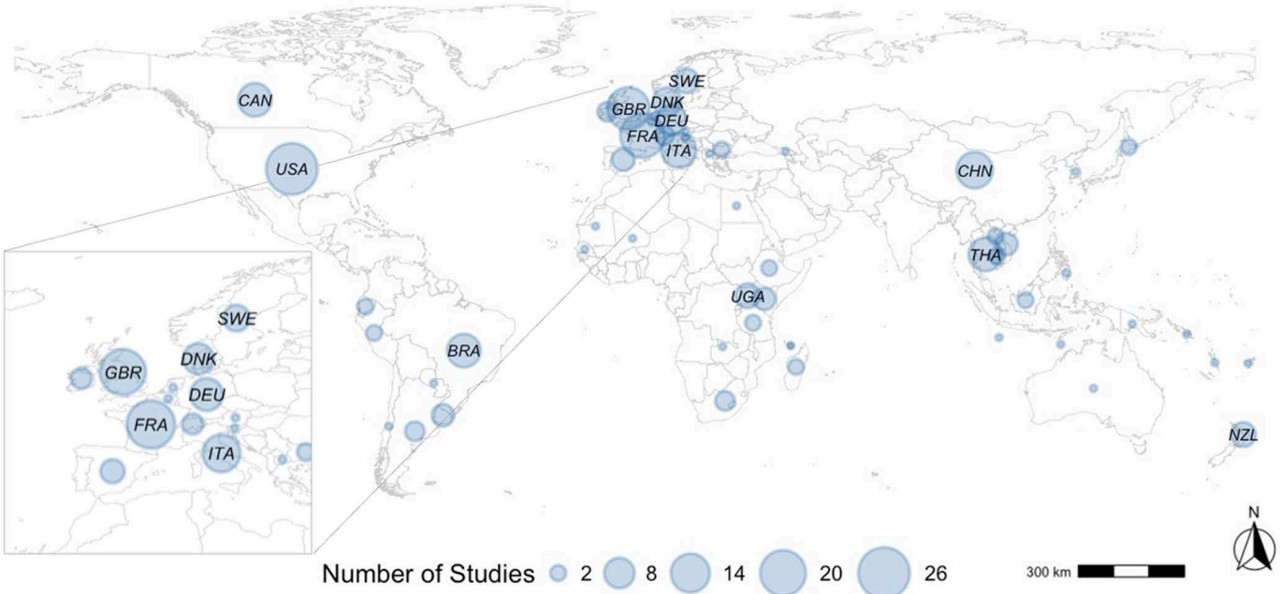

**Fig 3. Distribution of studies on livestock and poultry movement networks globally, up until 2023 (inclusive).** Circle sizes indicate the number of studies per country during this period. Countries with at least 5 studies are identified by their respective ISO 3166−1 alpha-3 codes (i.e., 3 letter country code). The figure is original, created by the authors the study's data.

**Table 1. Distribution of studies by species group for the top 5 most represented countries. The table includes the number (n) and percentage (%) of total studies conducted within each country, up until 2023. In some cases, the sum of percentages does not equate to 100%, since some studies included multiple species.**

| Country | Species | n (%) |
|---|---|---|
| 1.United States (USA) | Swine<br>Cattle | 17 (65.4)[1]<br>10 (38.5)[1] |
| 2.France (FRA) | Cattle<br>Swine<br>Other | 9 (47.4)<br>8 (42.1)<br>2 (10.5) |
| 3.Great Britain (GBR) | Cattle<br>Other | 12 (70.6)<br>5 (29.4) |
| 4.China (CHN) | Poultry<br>Swine | 7 (58.3%)<br>5 (41.7%) |
| 5.Italy (ITA) | Cattle<br>Other | 8 (72.7)<br>3 (7.3) |

*Note:*

[1]*One study from the USA included multiple species (swine, cattle, poultry and small ruminants).*

*These five countries collectively account for 41.9% of the 203 studies.*

Most of the studies (128 studies, 63.1%) have considered disease outcome(s) in their analysis. Of these, about 51.6% (66) conducted simulation scenarios, and within these, 56.1% (37) did not specify a specifically named disease. A total of 28 different pathogens or diseases were examined, with the most common being bovine tuberculosis (16, 7.7%), Foot and Mouth Disease (14, 6.7%), Avian Influenza viruses (13, 6.3%), and African Swine Fever (10, 4.8%). Only two studies (1%) included multiple disease conditions.

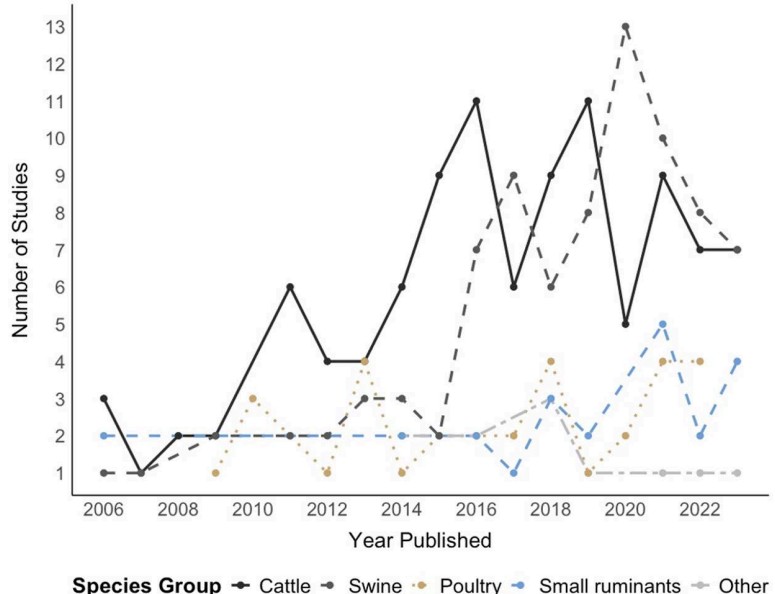

**Fig 4. Distribution of studies by species group over time from 2006 to 2023.** The "Other" species category includes less frequently investigated animals, namely buffalo, camelids, badgers, deer and ostriches, which were always explored alongside one of the other livestock or poultry groups.

The type of software used for network analysis has shown a distinct trend over time. Since 2013, R emerged as the dominant software tool for this type of analysis, while earlier studies opting for software like Pajek, Ucinet and Netdraw.

**Data extraction sources and exceptions.** The data extraction sources varied considerably by region and species, reflecting different data availability. Many studies (n = 86, 42.8%) sourced their data from national databases, particularly those carried in European countries, and only 8 databases were available electronically. In contrast, studies conducted in North America and Brazil typically relied on animal movement permits that were aggregated at the state or regional level (n = 46), with some using privately-owned data (n = 10) or voluntarily obtained data (n = 8). Notably, there was an increasing trend toward simulating data in the United States, as a response to the challenges posed by fragmented data and confidentiality concerns. For regions like China, African-countries, and low- and middle-income countries from Southeast Asia, survey-based approaches and snowball sampling techniques were more common (n = 43). Secondary data use was rare, occurring in only 2 studies, and international-level databases did not appear to exist.

Exceptions in data reporting were noted in the United States, where animal movements to slaughter plants are not required to be reported as these are considered "endpoints" and, therefore, unlikely to affect disease transmission. Additionally, only movements across state lines are required to be reported, leaving out more localized movements. In countries such as the US, but also Tanzania and Germany, exceptions also exist for specific movements to livestock markets or other approved facilities. Nevertheless, most studies did not report exclusions related to the movement data (n = 92, 87.6%).

**Characteristics of animal movement networks.** Fig 5 outlines key characteristics across all studies investigating livestock and poultry movement networks. Most studies (n = 148, 60.9%) utilized individual premises or animal holdings as nodes, while 29.1% used aggregated nodes, such as regions (districts, subdistricts, counties, provinces), with counties being the most frequent regional representation (n = 15, 27.3% of the regional representations). A smaller subset of studies (37, 18.2%) employed multiple node types, capturing more complex movement dynamics across varied scales. Specific

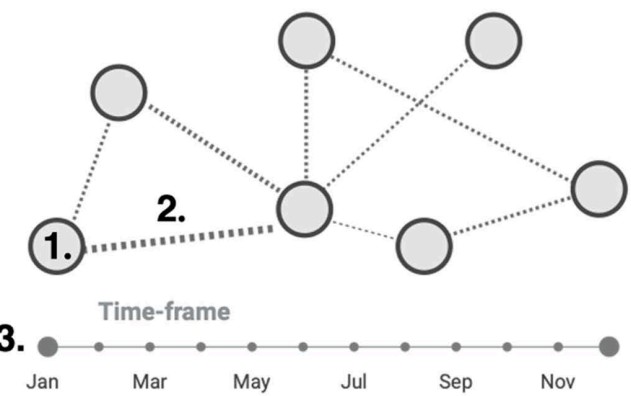

| Attribute | Frequency (n, %) | Notes |
|---|---|---|
| 1. Node type | Individual premises (148, 60.9%), Regions (55, 29.1%) | Counties were the most common regional node. |
| 2. Edge weight | No. of animals (68, 48.9%), No. of Movements (61, 43.9%) | Other weights include transport distance and average no. of animals moved (4, 2.9%, for each). |
| 3. Temporal scale | Static (78, 38.4%), Monthly (62, 28.8%), Annual (52, 25.6%), Daily (35, 17.2%), Weekly (32, 15.8%) | Finer temporal scales were less common. |

¹ The percentages reported relate to the total number of studies (n=203). Some attributes were considered in more than one study, leading to overlapping counts.

**Fig 5. Representation of network attributes in livestock and poultry movement network studies.** The diagram illustrates the components of network analysis – node types (1), edge weights (2), and temporal scales (3) – while the accompanying table provides a detailed summary of their respective frequencies and characteristics across studies.

nodes such as markets (n = 20, 8.2%) and traders (n = 4, 1.6%) were less common. Notably, almost all studies (n = 202, 99.5%) represented animal movements as directed networks, reflecting the origin and destination of each movement; only one study used an undirected approach [41]. Approximately 57.6% of studies incorporated weighted connections to quantify the intensity of movements between nodes. The two most common metrics for edge weights were the number of animals transported (n = 68, 48.9%) and the number of movements (n = 61, 43.9%). Additional metrics, though less frequently applied, included transport distance (n = 4, 2.9%), average number of animals moved weekly or monthly (n = 4, 2.9%), and the number of trucks or transport companies involved (n = 2, 1.4%).

Studies also varied in their consideration of temporal scales depending on the nature and objectives of the research (Figs 5–3). Temporal dynamics were addressed in 61.6% (n = 125) of the studies, typically by using monthly networks, with fewer studies adopting finer, daily or weekly temporal scales. Data periods ranged from a few months to multiple years, with some studies focusing on specific time frames when assessing specific disease outbreaks or seasonal movement patterns. Most commonly, studies considered one year of records (64, 31.5%), though three years or less than 6 months years of data were also frequent. Studies' length was largely dictated by data availability and research's regional or national scope. Similarly, the number of networks analyzed per study varied widely, from a single network to simulations of up to 950,000 networks. While the mean number of networks per study was 5504.7 (SD = 67990.1), this was skewed by simulation-based studies that often-constructed larger numbers of networks for robustness.

Few studies incorporated descriptive variables into their analysis. The majority type (n = 116, 57.1%) focused solely on the type of premises, which defined each node, while other descriptive factors were less commonly included. Only 36 (17.7%) included age of the animals, 28 (13.8%) accounted for the purpose of movement or reason for sale, 25 (12.3%) considered the sex of the animals, and 23 (11.3%) included breed or species type. Transport details, such as truck identification or transport companies, appeared in only 15 studies (7.3%), and specific variables like herd size, farm density, price, or weight, were included in fewer than 10%. Countries with more sophisticated data collection systems, such as European nations, often included more comprehensive descriptive variables, while regions with less established data infrastructure, such as Southeast Asia and parts of Africa, relied more on basic movement data.

## Underlying motivations

The qualitative analysis revealed five key themes that explain the underlying motivations for adopting network analysis approaches in studying livestock and poultry movement dynamics:

**Network characteristics and structural analysis.** Exploring network characteristics was the most prevalent motivation, identified in 73.4% of the studies (n = 144). The primary goal of this theme is to characterize the fundamental structure of animal movement networks [42,43], identifying critical nodes [44,45] and understanding the properties of the network nodes that influence disease dynamics [46]. Studies under this theme often aimed to classify premises based on network measures or explore the effects of temporality on trading patterns [19,47,48]. By understanding the structural aspects of animal movement, researchers could identify key nodes within the network that could be targeted for intervention. This is critical for designing more efficient disease surveillance and response strategies. The rationale behind this approach is that by characterizing network structure, veterinarians and policymakers can better understand which nodes are essential for disease spread and how to target interventions.

**Pathogen spread and epidemic modeling.** Approximately 35.0% of the studies (n = 71) focused on understanding pathogen transmission within animal movement networks. This theme combined several objectives, including simulating disease spread [49,50], determining potential epidemic size [51,52], and quantifying the role of network structure in facilitating disease dynamics [53,54]. Studies in this category aimed to model and quantify disease spread within animal movement networks [55], identify risk factors for transmission [46], and assess the potential for multi-host transmission [56,57]. By analyzing network structures and their effects on pathogen spread, researchers aimed to predict potential epidemic outbreaks and understand how animal movements contribute to disease dynamics. This theme reflected the importance of using network analysis to inform epidemic risk assessments and develop strategies for preventing large-scale disease outbreaks.

**Targeted control and surveillance measures.** Identified in 31.5% of the studies (n = 64), this theme emphasized the need for focused, network-based surveillance and control strategies. A significant portion of these studies examined the efficacy of various control strategies and identified specific nodes for targeted surveillance [58–67]. The focus was on pinpointing critical components within networks that could be leveraged for more effective disease mitigation. The rationale behind this motivation is that disease spread can be more efficiently controlled by focusing on high-risk nodes, which could be premises or movement pathways that are more likely to contribute to transmission. Targeted interventions informed by network analysis offer a more strategic and efficient approach to managing disease outbreaks and reducing overall disease burden.

**Retrospective analysis of outbreaks.** About 14.8% of the studies (n = 30) focused on understanding pathogen dynamics and network behavior during and after outbreaks. This theme included studies that analyzed the association between network features and disease positivity [21,68–70], as well as those investigating disease dynamics post-outbreak [71–74]. The key motivation here was to understand how disease spread within networks after an outbreak and to identify common features in network structure that correlate with disease presence. By analyzing these patterns, researchers aimed to refine control measures and improve future surveillance efforts. This theme highlighted

the importance of studying disease outbreaks retrospectively to inform ongoing and future disease management strategies.

**Network inference.** Also identified in 14.8% of the studies (n = 3), this theme focused on methodological advancements in network analysis. It included studies that utilized statistical models like Exponential Random Graph Models (ERGMs) to generate movement networks [75,76] and those evaluating methods to predict epidemics using partial data [77,78]. The underlying motivation of this theme was to overcome the challenges of incomplete or sparse data by developing innovative approaches to model animal movement networks. These studies aimed to improve the accuracy and reliability of network-based disease surveillance systems, particularly in contexts where complete data may not be available. By improving methods for data inference, researchers aimed to provide more robust predictions of disease spread and inform control measures.

### Strengths and limitations

This review identified both strengths and limitations in the application of network analysis to animal movement studies. While some aspects of these methodologies were considered advantages in certain contexts, they posed challenges in others. For instance, some studies referred the use of static networks as beneficial for simplifying analyses, but others have mentioned it could lead to overestimation of connectivity in dynamic systems. Table 2 summarizes these contrasting perspectives, highlighting how methodological choices might influence study outcomes and applicability.

## Discussion

To the best of our knowledge, this is the first scoping review to comprehensively explore the application of network analysis to livestock and poultry movement data, offering critical insights into its utility for disease surveillance and control.

The findings revealed a steady increase on the use of network-based approaches in veterinary preventive medicine since 2006, potentially following the 2001 FMD outbreak in the United Kingdom [73,79]. This event underscored the importance of animal movements in disease spread, driving advancements in traceability systems across countries and network-based methodologies for several species. Since then, these techniques have been extended to address a diverse range of pathogens, highlighting its versatility and expanding scope of network analysis in veterinary epidemiology.

The review identified five core themes underlying the motivations for adopting network analysis: network characteristics and structural analysis, pathogen spread and epidemic modeling, targeted control and surveillance measures, retrospective outbreak analysis, and network inference. These themes collectively highlight the flexibility of network analysis in addressing various aspects of disease dynamics, from identifying super-spreaders to evaluating and refining targeted interventions. The high prevalence of studies focusing on structural analysis reflects its foundational role in understanding disease spread, while the increasing use of network inference demonstrates advancements in addressing data limitations, particularly in regions with sparse or incomplete records.

Despite its growing utility, major differences exist across studies in terms of data granularity and accessibility, influenced by country or regional differences in infrastructure, regulations, and cultural practices. In several European countries and regions, national databases have facilitated detailed analyses, whereas in low- and middle-income countries, reliance on surveys and aggregated data limits the robustness of network models. Addressing these disparities would require investment in standardized, real-time data collection systems that capture comprehensive movement information across species and regions. Furthermore, the limited inclusion of descriptive variables in many studies, such as age, sex, and purpose of movement, can significantly limit our understanding of pathogen transmission patterns. Incorporating these variables, alongside economic and geographic data, could enhance the predictive power of network models, providing a more holistic understanding of disease spread and control needs. Notably, this review underscored the critical role of collaboration between public authorities, private stakeholders, and academic researchers [80]. Expanding such collaborative efforts, particularly in regions with fragmented data systems would help with improvement of the scope and accuracy of disease surveillance.

**Table 2. Reported strengths versus limitations in applying social network analysis to livestock and poultry movement networks.**

| Application Component | Strengths | Limitations |
|---|---|---|
| **Network Structure** | | |
| Static Networks | ◇ Provide valuable metrics for epidemic size; simplify analysis of contact structures. | ◇ May overestimate connectivity, inflating disease spread perceptions; static networks risk spurious inferences in rapidly changing environments. |
| Weighted Networks | ◇ Reflect variations in transmission likelihoods, improving accuracy in identifying high-risk nodes. | ◇ Requires detailed data for weight calculation. |
| Data Granularity and Completeness | ◇ Tracing individual animals offers detailed insights into disease dynamics at a micro-level. | ◇ High data granularity requires advanced data management (impractical for large-scale studies); tracking at the animal-level can be cost-prohibitive. |
| Network inference | ◇ QAP1 methods and weighted inference enable robust analysis of partially observed networks, improving disease modeling accuracy. | ◇ Inference from partial networks may introduce biases; QAP use requires caution due to multi-collinearity; inaccurate predictions if not properly validated. |
| Network Dynamics and Temporality | ◇ ERGMs2 and temporal models can simulate realistic contact probabilities | ◇ Lack of real-time, longitudinal data limits the applicability of temporal models. |
| **Network-targeted Mitigation Strategies** | | |
| Compartmentalization of communities | ◇ Enables targeted surveillance by identifying at-risk communities; supports zoning and regionalization strategies. | ◇ Requires further investigation of other disease pathways; required validation for diverse pathogen transmission routes. |
| Special considerations for small producers | ◇ Recognizes unique movement patterns and biosecurity needs for small herds. | ◇ Lack of resources and limited disease awareness in these settings may lead to non-compliance |
| Biosecurity Protocols | ◇ Allows the improvement of biosecurity targeted at high-risk 'super-spreaders' and vehicles. | ◇ Low farm-compliance with biosecurity standards; logistical challenges in enforcing strict cleaning measures for transport vehicles across regions. |
| Market-based surveillance | ◇ Markets act as disease transmission hotspots; targeted surveillance at high-centrality markets could efficiently manage disease spread. | ◇ Obtaining data from these facilities is challenging; Limited infrastructure in certain markets may hinder collection of such data. |
| **Data Collection and Quality** | | |
| Developing Country Challenges | ◇ Simple surveys can help track animal movement coverage, especially for small, backyard farms. | ◇ Limited veterinary resources and infrastructure in rural areas complicate consistent data collection and intervention strategies. |
| Data Awareness and Traceability | ◇ Emphasizes the need for accurate, consistent movement data; raises awareness among stakeholders about data importance for disease control. | ◇ Inconsistent data reporting and participation by farms; lack of real-time data hampers timely interventions; difficulties in incentivizing compliance with recording standards, particularly among private sector stakeholders. |
| **Interdisciplinary and Collaborative Networks** | | |
| Expanded Network Complexity | ◇ Including additional transmission routes (e.g., fomites, vehicles, personnel) provides a more complete picture of disease spread dynamics. | ◇ Incorporating multiple routes increases model complexity and data requirements; lack of available data on non-animal transmission vectors complicates network analysis. |
| Cross-Sector Collaboration | ◇ Collaboration with veterinarians and researchers accelerates response times and improves network understanding; public-private partnerships provide a great opportunity for data standardization. | ◇ Differences in priorities and resources across stakeholders can impede coordinated action; trust and data sharing may be limited among independent producers or markets. |

[1]QAP – Quadratic Assignment Procedure (QAP)

[2]ERGMs – Exponential Random Graph Models

The relevance of our findings extends to key groups, including veterinarians, epidemiologists, policymakers, and livestock producers. Across the reviewed studies, identifying critical nodes within animal movement networks was frequently proposed as a means to inform targeted biosecurity measures for disease prevention and optimize resource allocation during outbreaks. Incorporating weighted connections that reflect movement intensity was also noted to provide a more

nuanced understanding of transmission risks, supporting the design of more effective prevention and control strategies. Based on the findings, we recommend future research to clearly define nodes in alignment with the relevant epidemiological units. While premises offer more granular resolution, regional-level nodes are often the only available choice and may suffice when disease mitigation strategies are implemented at broader administrative levels. The selection of network metrics should also be guided by the specific research question, as different metrics offer distinct information. These choices should be tailored to the production system, pathogen characteristics, and the temporal and structural properties of the network of interest.

Despite the demonstrated utility of network analysis, several gaps remain. To start, the absence of international-level databases hinders cross-country comparisons and the development of global disease control strategies. Establishing such databases, with standardized reporting protocols, could enable more comprehensive analyses and facilitate collaboration among nations. Moreover, only about 8 country-wide databases were reported as available electronically, limiting data accessibility. The use of data from previous studies was also rare, with only two studies incorporating such data, and should be encouraged. Additionally, the underrepresentation of specific nodes, such as markets and traders, pointed to a need for more inclusive data collection efforts. Markets, in particular, have been linked to disease transmission risks – some studies demonstrating direct associations with outbreaks [61,64,81], while others highlight their structural role and centrality within movement networks as indicative of potential risk [82,83] – yet data from these settings remain scarce. Similarly, the underrepresentation of small ruminant' networks may reflect not only data limitations but also the inherent challenges of tracking animals in more extensive, pastoral or smallholder settings, where centralized data collection is often not feasible. Overcoming challenges related to data confidentiality and logistical barriers is critical for capturing the full complexity of animal movement networks. Most studies also relied on static or largely aggregated networks, which may oversimplify the dynamic nature of animal movements. Developing models that account for temporal variations [75,84] and multiple transmission pathways [85], including indirect routes such as vehicles and personnel, is a promising avenue for future research. Moreover, it is also important not to exclude other relevant transmission routes, such as fomites, wildlife, or environmental reservoirs, that may contribute to pathogen dissemination. Such advancements could provide more accurate risk assessments and inform real-time interventions. Finally, while many studies proposed network-informed interventions, we must note that few have demonstrated their implementation in real-world scenarios. Most examples in the literature involved scenarios based on network-informed transmission models which, while valuable, are inherently limited by the assumptions they rely upon and the quality of input data. The challenge of translating these insights into practices reflects both logistical difficulties in outbreak settings and the complexity of coordinating policy shifts based on predictive tools. Nonetheless, the ability to incorporate detailed movement data into disease modeling remains a promising tool.

This scoping review was meant to offer a comprehensive, yet systematic, overview of existing evidence. Given its descriptive nature, risk of bias for scoping reviews is typically not assessed neither is a considerable concern. Nonetheless, to minimize bias in evidence collection, this study endeavored to gather a wide array of eligible studies, spanning different languages, and incorporated select grey literature, recognizing its potential significance as valuable evidence. While our search strategy was designed to be as inclusive as possible, we acknowledge that there is always a possibility of missing some studies, especially grey literature and non-indexed reports. As seen in the PRISMA flow diagram (Fig 1), only 4 studies were excluded based on language; however, some relevant studies might not have been captured due to the reliance on search terms in English, potentially limiting the global scope of the review. Additionally, the inclusion of grey literature, though valuable for capturing diverse perspectives, may introduce variability in methodological quality. In this review, only a small portion of studies (n = 7, 3.4%) originated from grey literature sources, including mostly PhD, post-graduation, and master's theses, along with one conference paper. Another limitation was that only one person conducted "title and abstract" screening, which may have introduced the risk of oversight or individual bias. Despite these limitations, the systematic approach and adherence to PRISMA-ScR guidelines provide confidence in the comprehensiveness of the evidence synthetized in this review.

## Conclusions

In conclusion, network analysis has proven a powerful tool in identifying at-risk nodes or groups within livestock and poultry movement networks, offering promising theoretical insights for targeted surveillance and biosecurity strategies that may outperform traditional approaches. Despite its strengths, challenges remain, particularly regarding data granularity, temporal dynamics, and the inclusion of multiple transmission routes. Looking forward, research opportunities are abundant. Expanding data collection efforts to incorporate real-time, multi-species, and multi-pathway data will be essential to capture the complexity of animal movement networks fully. Integrating network analysis with economic and geographic information systems could further enhance pathogen spread understanding and control strategies.

There is also an urgent need to foster cooperation between public authorities and private stakeholders to ensure consistent, reliable data reporting across animal production industries. Addressing these gaps will not only strengthen network analysis methodologies but also contribute to more resilient and responsive disease control strategies in the livestock and poultry sectors.

## Supporting information

**S1 File. PRIMA-P Check-list.** The PRISMA 2020 statement: an updated guideline for reporting systematic reviews. (DOCX)

## Acknowledgments

We sincerely thank JP, one of the co-authors, for her invaluable support throughout this systematic review. We also extend our gratitude to AA for her encouragement in undertaking this comprehensive review.

## Author contributions

**Conceptualization:** Sara C. Sequeira, Natalie Sebunia, Andréia G. Arruda, Jessica R. Page.

**Data curation:** Sara C. Sequeira, Natalie Sebunia.

**Formal analysis:** Sara C. Sequeira.

**Investigation:** Sara C. Sequeira.

**Methodology:** Sara C. Sequeira, Andréia G. Arruda, Jessica R. Page, Taiwo Lasisi.

**Project administration:** Sara C. Sequeira.

**Supervision:** Andréia G. Arruda.

**Writing – original draft:** Sara C. Sequeira.

**Writing – review & editing:** Sara C. Sequeira, Natalie Sebunia, Andréia G. Arruda, Greg Habing, Jessica R. Page, Taiwo Lasisi.

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
