## [Decision Letter · Decision Letter 0]

PONE-D-25-08168A systematic scoping review analysis: How can livestock and poultry movement networks inform disease surveillance and control at the global scale?PLOS ONE

Dear Dr. Sequeira,

Thank you for submitting your manuscript to PLOS ONE. After careful consideration, we feel that it has merit but does not fully meet PLOS ONE’s publication criteria as it currently stands. Therefore, we invite you to submit a revised version of the manuscript that addresses the points raised during the review process.

**ACADEMIC EDITOR: ** Dear authors, you should make a minor revision to your manuscript depending on the reviewers' suggestion for change. Please submit your revised version, responding to each comments raised by the reviewer's.

If applicable, we recommend that you deposit your laboratory protocols in protocols.io to enhance the reproducibility of your results. Protocols.io assigns your protocol its own identifier (DOI) so that it can be cited independently in the future. For instructions see: https://journals.plos.org/plosone/s/submission-guidelines#loc-laboratory-protocols. Additionally, PLOS ONE offers an option for publishing peer-reviewed Lab Protocol articles, which describe protocols hosted on protocols.io. Read more information on sharing protocols at https://plos.org/protocols?utm_medium=editorial-email&utm_source=authorletters&utm_campaign=protocols .

We look forward to receiving your revised manuscript.

Kind regards,

Nussieba A. Osman, Dr. Med. Vet.

Academic Editor

PLOS ONE

Journal Requirements:

2. Thank you for stating the following financial disclosure: [There is no specific funding for the scoping review hereby presented but the main authors' research is supported by the Agriculture and Food Research Initiative Competitive Grant no. 2022-68015-36628, awarded by the United States Department of Agriculture.].

4. We note that Figure 3 in your submission contain [map/satellite] images which may be copyrighted. All PLOS content is published under the Creative Commons Attribution License (CC BY 4.0), which means that the manuscript, images, and Supporting Information files will be freely available online, and any third party is permitted to access, download, copy, distribute, and use these materials in any way, even commercially, with proper attribution. For these reasons, we cannot publish previously copyrighted maps or satellite images created using proprietary data, such as Google software (Google Maps, Street View, and Earth). For more information, see our copyright guidelines: http://journals.plos.org/plosone/s/licenses-and-copyright.

5. Please include captions for your Supporting Information files at the end of your manuscript, and update any in-text citations to match accordingly. Please see our Supporting Information guidelines for more information: http://journals.plos.org/plosone/s/supporting-information .

Reviewers' comments:

Reviewer's Responses to Questions

**Comments to the Author**

1. Is the manuscript technically sound, and do the data support the conclusions?

Reviewer #1: Yes

Reviewer #2: Yes

2. Has the statistical analysis been performed appropriately and rigorously? 

Reviewer #1: Yes

Reviewer #2: Yes

3. Have the authors made all data underlying the findings in their manuscript fully available?

Reviewer #1: Yes

Reviewer #2: Yes

4. Is the manuscript presented in an intelligible fashion and written in standard English?

Reviewer #1: Yes

Reviewer #2: Yes

5. Review Comments to the Author

Reviewer #1: This article is very interesting as it is providing a nice overview of the application of network analysis in veterinary epidemiology, the strengths and limitations that have been identified in this area. However, regarding the claimed benefits of identifying key nodes to guide risk-based surveillance for instance, or to tailor control measures, my question is: have these been demonstrated to date? It seems that the implementation of the outcomes of these studies with associated positive results has yet to be demonstrated. Perhaps this is due to the limitations that have been identified by the studies that are reported in the present paper. But it would add to the manuscript to have a brief discussion about this. Theoretically, it makes sense, but I have yet to see a publication showing its real-world application.

- Although there is mention that FMD 2001 likely led to a number of network analysis studies, I believe there are other factors that have contributed to the observed body of literature in this area. It would have been nice to show on Figure 4, or in the discussion, associated major disease events or changes in control policies, or data systems becoming online… that could explain the rise in network analysis studies:

o ASF in Eastern Europe as of 2014 and ASF in China in August 2018 – these events likely explain the observed rise in studies pertaining to swine;

o Cattle passport system with electronic database in the EU since 2000;

o Of course FMD in 2001 but also bluetongue perhaps in Europe since 2006-2010?

o HPAI in poultry; are peaks in studies associated with peaks in outbreaks globally?

o Ongoing tuberculosis control in various countries?

o Spread of PPR in Europe since around 2016?

Specific comments :

Line 66 : Reference 8 is not a good example of “robust livestock and poultry traceability systems worldwide”, but is a reference that mentions good examples of such systems. Suggest you list references that are direct examples of such robust systems.

Line 80: “… not only used to trigger hot spots…” . I am not sure what this means. The way that a network is organized as a result of the specific production system, may lead to the emergence of hot spots or super-spreaders that may be involved in pathogen transmission and their identification in a specific network may inform surveillance and control for targeted, risk-based, surveillance. I believe this is what you are trying to say? As I don’t know how the dynamics of the network could trigger hot spots…

Line 85: animal transportation networks. There are animal movement networks that are not based on transportation. Transhumance, pastoral-type of systems where animals are moved on foot, sometimes across borders etc… Were these included in the scope, and if they were, perhaps there were none published? In the discussion, I think it is important to note if the majority of papers come from countries with similar types of production systems that are more apt to have movement databases than others. Regardless of the means to develop such systems, the type of production just doesn’t allow their implementation, i.e., small ruminants, which can explain why there are less papers published. Such analysis in thinking about why you have the descriptive statistics that you have would have brought an interesting discussion to the paper.

Line 86-87: sentence doesn’t read well. Suggest: “One available piece of work focused on a review of the different network-based modelling frameworks used to represent animal movements (24).”

Line 95: not sure what you mean by transportation systems. Should be defined or explained as this is not a criteria that you used in the analysis.

Line 129: animal movement or transportation data. Again, please explain so the reader can understand what you mean, i.e., how they differ.

Line 199: animal movement studies is used here. Please be consistent.

Line 251: could have the reference here since mentioning a specific article?

Table 1: would help the reader if added lines separating the countries as it was difficult to follow.

Line 322: It would be of added value to know if the papers used directed or undirected networks in their studies. The use of the term “edge” in the manuscript can create confusion. The more precise term to use in directed networks is “arc” to represent the directional movement from one node to another. I suspect that the vast majority of the studies that you considered are based on directed network, but this should be mentioned in the descriptive statistics.

- This raises one question. A number of papers that have been published explored different network analysis measures to determine which could be the more useful to meet certain objectives: predict epidemic size, construct network structures on which to model disease spread etc.. It seems this theme was not picked up in your study where you focus on network structure, not network measures. There has been a body of work looking at applying social network analysis measures and relating them to veterinary epidemiology and it seems this is not reflected in your study. Perhaps this needs to be discussed?

Line 380: replace “animal networks” with “animal movement networks”

Table 2:

- Data collection and quality: in the discussion, there should be mention of small ruminants where sure, it can be a matter of veterinary resources and infrastructure that complicate the ability to obtain information on this population, but also, it is the structure of that sector that also makes it very difficult to track? Yet, they are extremely important components of the global spread of diseases, think FMD, PPR etc… This should be covered in the discussion, there are certainly technological challenges at gaining traceability for this population.

Line 467: in this discussion it must be added that animal movements, while important, are one pathway for disease transmission and that others such as fomites, wildlife, etc… are important as well. By focusing only on animal movement, we do not have full comprehension of the network structure.

Line 467-479: I don’t see the point of this example which is not discussed in the rest of the manuscript. Reference 75 can be used to represent public private and academic partnerships.

Line 480-484: these benefits keep being mentioned and are theoretical at this time. Are there any papers that actually have shown how implementing the findings from animal movement studies have benefited either early detection or control of diseases?

Line 524 – 527: I think that the first part of this sentence is true (identifying potentially high-risk components – though components in network analysis has a meaning, perhaps find another word), theoretically, but the second part of the sentence has not been demonstrated in animal movement network analysis studies (underscoring the importance of targeted surveillance and biosecurity measures, as opposed to other approaches).

The limitations that researchers identify, especially in terms of completeness of data to really completely understand and identify high-risk nodes mean that there has not been yet implementation of policies, surveillance and control measures solely relying on animal movement network findings. Identification is not the same as modelling disease spread as well.

Biosecurity helps all nodes, not just highly connected ones and I would doubt that any veterinary services around the world would target biosecurity measures only at high-risk nodes derived from incomplete networks. Until complete networks are available, I believe all this work remains informative and theoretical, without implementation. As a result, the language should be softened in this paper.

Reviewer #2: This is a well-presented submission that addresses an important topic in animal health research. The authors are to be commended for producing an academically sound and practical review of an analytical tool used to understand the ecological and preventive components of animal diseases. The format and comprehensive detail of the scoping review are commendable. I wish more previously published manuscripts had followed a similar structure for improved clarity and impact. I have a few minor comments and suggestions that should not be viewed as obstacles to the acceptance of this submission:

1. Although the authors appropriately acknowledged the limitations of network analysis in their conclusions, I encourage them to include a few sentences discussing the limitations of the scoping review itself in identifying all relevant prior applications of network analysis. For example, I was unable to find references to two earlier technical reports on animal movement network analysis conducted for the European Union and Denmark. It appears that the authors may have been constrained by the availability of sources in the databases they accessed.

2. Based on their findings, the authors might consider adding a few sentences offering recommendations on when and how to apply network analysis, including any requirements or considerations. Such guidance would be valuable for future users and decision-makers, especially if supported by evidence from prior work.

I wish the authors continued success in maintaining this high standard of research to support better disease management in both human and animal populations.

6. PLOS authors have the option to publish the peer review history of their article (what does this mean? ). If published, this will include your full peer review and any attached files.

**Do you want your identity to be public for this peer review?** For information about this choice, including consent withdrawal, please see our Privacy Policy .

Reviewer #1: No

Reviewer #2: **Yes: ** Mo D Salman

---

## [Author Response · Author response to Decision Letter 1]

29 Apr 2025

All comments were answered and can be consulted in the provided document "Response to Reviewers". Thank you again for your valuable reviews.

---

## [Decision Letter · Decision Letter 1]

PONE-D-25-08168R1A systematic scoping review and thematic analysis: How can livestock and poultry movement networks inform disease surveillance and control at the global scale?PLOS ONE

Dear Dr. Sequeira,

Thank you for submitting your manuscript to PLOS ONE. After careful consideration, we feel that it has merit but does not fully meet PLOS ONE’s publication criteria as it currently stands. Therefore, we invite you to submit a revised version of the manuscript that addresses the points raised during the review process.

**ACADEMIC EDITOR: ** Dear Authors, Thank you for modifying the manuscript. After an initial revision, reviewers advised that certain points still require revision or amendment. Please carefully revise your paper and respond to all of the reviewers' comments.

We look forward to receiving your revised manuscript.

Kind regards,

Nussieba A. Osman, Dr. Med. Vet.

Academic Editor

PLOS ONE

Journal Requirements:

Reviewers' comments:

Reviewer's Responses to Questions

**Comments to the Author**

1. If the authors have adequately addressed your comments raised in a previous round of review and you feel that this manuscript is now acceptable for publication, you may indicate that here to bypass the “Comments to the Author” section, enter your conflict of interest statement in the “Confidential to Editor” section, and submit your "Accept" recommendation.

Reviewer #1: All comments have been addressed

Reviewer #2: (No Response)

Reviewer #3: All comments have been addressed

2. Is the manuscript technically sound, and do the data support the conclusions?

Reviewer #1: Yes

Reviewer #2: Yes

Reviewer #3: Yes

3. Has the statistical analysis been performed appropriately and rigorously? 

Reviewer #1: Yes

Reviewer #2: Yes

Reviewer #3: Yes

4. Have the authors made all data underlying the findings in their manuscript fully available?

Reviewer #1: Yes

Reviewer #2: Yes

Reviewer #3: Yes

5. Is the manuscript presented in an intelligible fashion and written in standard English?

Reviewer #1: Yes

Reviewer #2: Yes

Reviewer #3: Yes

6. Review Comments to the Author

Reviewer #1: Dear Author,

Thank you for your review of the comments I put forward. I am in agreement with your answers and your modifications in the manuscript that has improved as a result. Just a few minor things:

line 86 and 94, check again the use of transportation instead of movement.

Line 203 , perhaps it is better to say application of network analysis to (instead of in) animal movement data?

Line 515: are recognized as potential (word added) hotspots? In your references, perhaps two demonstrate the actual risk whereas others suggest based on network?

Reviewer #2: Congratulations for good submission .

Reviewer #3: The article is very interesting and well-written. The authors conducted rigorous, extensive, and interesting work that needs to be shared. There are only a few minor points that I wish the authors would address.

In the abstract:

The sentence "The increasing threat of emerging infectious diseases affecting animal and human populations has prompted a critical examination of their origins and contact patterns driving disease transmission" could be misleading. " could be misleading. I usually associate transmission with a local event (a susceptible individual being infected by another) and use the term "propagation" when discussing epidemics that appear in different subpopulations. Similarly, contact patterns make me think of individual habits. If you could rephrase to focus more on the interaction between multiple locations, it would greatly improve readability.

The same goes for the sentence, "Five themes emerged: network structure (73.4%), epidemic modeling (35.0%), targeted control (31.5%), outbreak analysis (14.8%), and network inference." could be misleading. I suppose there is a certain degree of overlap among the articles. Perhaps it would be better not to show percentages.

In Section 3.2.1,

It would be nice to stratify Figure 3A by continent or economic area to show whether works in these areas are becoming more frequent.

Similarly, it would be nice to present a table similar to Table 3, organized by geographical continent or economic area.

7. PLOS authors have the option to publish the peer review history of their article (what does this mean? ). If published, this will include your full peer review and any attached files.

**Do you want your identity to be public for this peer review?** For information about this choice, including consent withdrawal, please see our Privacy Policy .

Reviewer #1: No

Reviewer #2: No

Reviewer #3: **Yes: ** Andrea Apolloni

---

## [Author Response · Author response to Decision Letter 2]

16 Jun 2025

All responses were attached in "Response to Reviewers R2" document. We appreciate all the valuable feedback that improved the quality of our research study!

---

## [Editor Report · Decision Letter 2]

A systematic scoping review and thematic analysis: How can livestock and poultry movement networks inform disease surveillance and control at the global scale?

PONE-D-25-08168R2

Dear Dr. Sequeira,

We’re pleased to inform you that your manuscript has been judged scientifically suitable for publication and will be formally accepted for publication once it meets all outstanding technical requirements.

Kind regards,

Nussieba A. Osman, Dr. Med. Vet.

Academic Editor

PLOS ONE
---

## [Editor Report · Acceptance letter]

PONE-D-25-08168R2

PLOS ONE

Dear Dr. Sequeira,

I'm pleased to inform you that your manuscript has been deemed suitable for publication in PLOS ONE. Congratulations! Your manuscript is now being handed over to our production team.

Kind regards,

on behalf of

Dr. Nussieba A. Osman

Academic Editor

PLOS ONE